# Optically induced charge-transfer in donor-acceptor-substituted *p*- and *m*- C$_2$B$_{10}$H$_{12}$ carboranes

Lin Wu[1], Marco Holzapfel[2], Alexander Schmiedel[2], Fuwei Peng[1], Michael Moos[2], Paul Mentzel[2], Junqing Shi[1], Thomas Neubert[3], Rüdiger Bertermann[3], Maik Finze ◉[3], Mark A. Fox[4], Christoph Lambert ◉[2] ✉ & Lei Ji ◉[1] ✉

Icosahedral carboranes, C$_2$B$_{10}$H$_{12}$, have long been considered to be aromatic but the extent of conjugation between these clusters and their substituents is still being debated. *m*- and *p*-Carboranes are compared with *m*- and *p*-phenylenes as conjugated bridges in optical functional chromophores with a donor and an acceptor as substituents here. The absorption and fluorescence data for both carboranes from experimental techniques (including femtosecond transient absorption, time-resolved fluorescence and broadband fluorescence upconversion) show that the absorption and emission processes involve strong intramolecular charge transfer between the donor and acceptor substituents via the carborane cluster. From quantum chemical calculations on these carborane systems, the charge transfer process depends on the relative torsional angles of the donor and acceptor groups where an overlap between the two frontier orbitals exists in the bridging carborane cluster.

Carborane derivatives such as the three *closo*-C$_2$B$_{10}$H$_{12}$ isomers (Fig. 1) and their derivatives possess unique molecular and electronic structures[1], which have been exploited in various applications such as in uranyl capture[2,3], in medical chemistry[4–14], and as luminescent molecules[15–25]. The skeleton of the carborane cluster is formed by 3c-2e bonds and may be considered as a three-dimensional (3D) aromatic system[26]. Logically, the cage may act as a conjugated bridge with electron delocalization and charge transfer as found in a phenylene bridge within two-dimensional (2D) π-systems in optoelectronic functional chromophores. However, in *o*-carboranes with aromatic substituents at the cluster carbon atoms, the overlap between the frontier molecular orbitals (FMOs) of the planar ring and the cage is found to be negligible thus the carborane is considered as a spacer here[27–33]. Similarly, *m*- and *p*-carboranes behave as spacers due to a lack of FMO overlap according to studies on carborane-bridged luminescent molecules[21,34–39] and mixed-valence systems[40–44].

While 2D aromatic bridges are effective in intramolecular charge transfer (ICT) processes, experimental studies of charge transfer processes in 3D carborane bridges remain unexplored. Wade and co-workers revealed that the absorption spectra of donor-acceptor compounds containing phenylene-*p*-carborane-phenylene bridges are red-shifted. They disclosed that the interaction between carborane and phenylene is antibonding in the HOMO but in-phase in the LUMO[45,46]. A strong electron acceptor at a carbon atom in a carborane may unlock the potential of carboranes as effective conjugated bridges. This idea led us to design donor-acceptor luminophores with *p*- and *m*-carborane bridges, **DA-*p*Carb** and **DA-*m*Carb**, where the donor is a triarylamine substituent and the acceptor is a dimesitylboron (BMes$_2$) substituent (Fig. 1). In the analogous *o*-carborane system, the sterics of the bulky substituents present would affect the electronic processes therefore this was not examined here. The phenylene-bridged analogues, **DA-*p*Benz** and **DA-*m*Benz**, were

[1]Frontiers Science Center for Flexible Electronics (FSCFE), Shaanxi Institute of Flexible Electronics (SIFE) & Shaanxi Institute of Biomedical Materials and Engineering (SIBME), Northwestern Polytechnical University (NPU), 127 West Youyi Road, Xi'an 710072, China. [2]Institut für Organische Chemie, Julius-Maximilians-Universität Würzburg, Am Hubland, 97074 Würzburg, Germany. [3]Institut für Anorganische Chemie, Julius-Maximilians-Universität Würzburg, Am Hubland, 97074 Würzburg, Germany. [4]Department of Chemistry, University of Durham, South Road, Durham DH1 3LE, UK. ✉e-mail: christoph.lambert@uni-wuerzburg.de; iamlji@nwpu.edu.cn

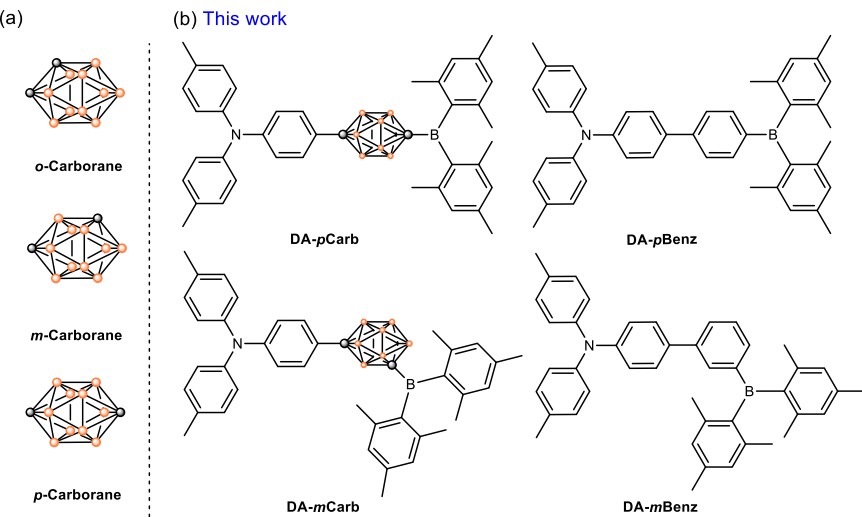

**Fig. 1 | Relevant structures in this work. a** Three isomers of *closo*-C$_2$B$_{10}$H$_{12}$ (carborane); **b** Structures of **DA-*p*Carb, DA-*m*Carb, DA-*p*Benz**, and **DA-*m*Benz** with distances between nitrogen and boron atoms at 10.6, 9.2, 10.2 and 9.0 Å, respectively.

also synthesized for comparison here. The empty $p_z$ orbital at boron makes the BMes$_2$ group a strong π-acceptor and the mesityl groups provide good stability by steric shielding[47–56]. Triarylamine is used as the electron donor due to its strong π-electron-donating ability and the low reorganization energy of the amine group supports rapid charge transfer processes[51,57,58]. The synthesis of all compounds are summarized in the SI (Supplementary Fig. 1–4). Our results demonstrate optically induced intramolecular charge transfer (ICT) through carboranes and confirm the conjugation between the cage and the π-system.

## Results

### Electrochemistry

To evaluate the donor and acceptor properties of the substituents attached to either carborane or phenylene bridge, cyclic voltammetry for all compounds were measured in THF for reduction and DCM for oxidation (Supplementary Fig. 5, 6). Similar reversible reduction and oxidation potentials against ferrocene/ferrocenium were found for all donor-acceptor systems (dyads) and are listed in Table 1 (see also Supplementary Fig. 6).

The oxidation potentials show that the carborane withdraws electron density from the triarylamine, thus the oxidation potential is increased by 120 mV in **DA-*p*Carb** and by 150 mV in **DA-*m*Carb** compared to tris(*p*-tolyl)amine ($E_{ox}$ = +0.34 in DCM)[59]. In contrast, the phenylene-bridged compounds retain similar oxidation potentials to tris(*p*-tolyl)amine. On the other hand, the *p*-carborane substitution leads to a more negative reduction potential than trimesitylborane (Mes$_3$B, $E_{red}$ = -2.43 V) by 60 mV, while the *m*-phenylene substitution is more negative than the *m*-carborane substitution. The band gaps based on the potentials are 22600 cm$^{-1}$ for **DA-*p*Benz**, 23700 cm$^{-1}$ for **DA-*p*Carb**, 23400 cm$^{-1}$ for **DA-*m*Benz**, and 23400 cm$^{-1}$ for **DA-*m*Carb** (Supplementary Fig. 5). These values show that the electronic

communication between the donor and the acceptor is stronger in the *para*-phenylene dyad than in the *meta*-phenylene analog. However, both carborane dyads display a similar band gap to the *meta*-phenylene dyad, indicating a similar degree of donor-acceptor interactions in these compounds.

### Absorption and fluorescence spectra

Evidence of the donor-acceptor behavior of the dyads was provided by absorption and emission spectra in solvents of different polarities (Fig. 2, Supplementary Fig. 7–13, Table 2). In hexane, **DA-*p*Carb** has a broad absorption band at 300 nm with a shoulder at 341 nm (Fig. 2a). Deconvolution analysis of the bands gave two Gaussian bands at around 29200 cm$^{-1}$ and 33000 cm$^{-1}$ (Supplementary Fig. 8–11), respectively. While the intense band at 33000 cm$^{-1}$ of **DA-*p*Carb** corresponds to a superposition of the localized (LE) transitions of the triarylamine and the BMes$_2$ group[60,61], the weaker one at 29200 cm$^{-1}$ is assigned the intramolecular charge transfer (ICT) between the amine and BMes$_2$ moieties through the *p*-carborane cage. The squared transition dipole moment (= dipole strength) of the ICT transition in hexane for **DA-*p*Carb** determined by integrating the ICT band in the absorption spectra is 14.7 D$^2$ (see Table 2)[62–64], which is about half that of the phenylene bridged **DA-*p*Benz** (31.8 D$^2$) where the ICT transition is visible as a separated absorption band at 26,000 cm$^{-1}$ in hexane (Supplementary Fig. 9). In both compounds, the observed ICT bands in the absorption spectra are little changed when going to a more polar solvent (Supplementary Fig. 12) indicating a small ground state dipole moment[46]. The 00-energy of the ICT band was estimated by the intersection of a tangent at the low-energy flank with the abscissa where the values of 27,400 cm$^{-1}$ for **DA-*p*Carb** and 23,900 cm$^{-1}$ for **DA-*p*Benz** were obtained. The absorption spectrum for **DA-*m*Carb** has one asymmetric broad band (Fig. 2a) where deconvolution analysis showed a strong LE band (32,700 cm$^{-1}$) and a very weak ICT band (29,200 cm$^{-1}$) (Supplementary Fig. 10) implying that the ICT transition in **DA-*m*Carb** is only weakly allowed. The deconvolution of the absorption spectrum in **DA-*m*Benz** also resulted in two intense bands, suggesting that the charge transfer through the *meta*-position of the phenylene ring is stronger than in the carborane analogue. However, we stress that the deconvolution of the absorption bands of the carborane and phenylene bridged *meta*-dyads is to some extent arbitrary. The similar ICT transition energies of **DA-*m*Benz** and the two carborane dyads are in agreement with similar band gaps measured by cyclic voltammetry for these compounds. The corresponding energies measured for **DA-*p*Benz** are significantly smaller.

**Table 1 | Electrochemistry data. Reduction and oxidation potentials of all donor-acceptor compounds measured by cyclic voltammetry vs. ferrocene/ferrocenium at 100 mV s$^{-1}$**

|                  | $E_{red}$ /V in THF | $E_{ox}$ /V in DCM |
|------------------|---------------------|--------------------|
| **DA-*p*Carb**   | −2.488              | +0.455             |
| **DA-*p*Benz**   | −2.429              | +0.377             |
| **DA-*m*Carb**   | −2.433              | +0.472             |
| **DA-*m*Benz**   | −2.538              | +0.362             |

**(a)**

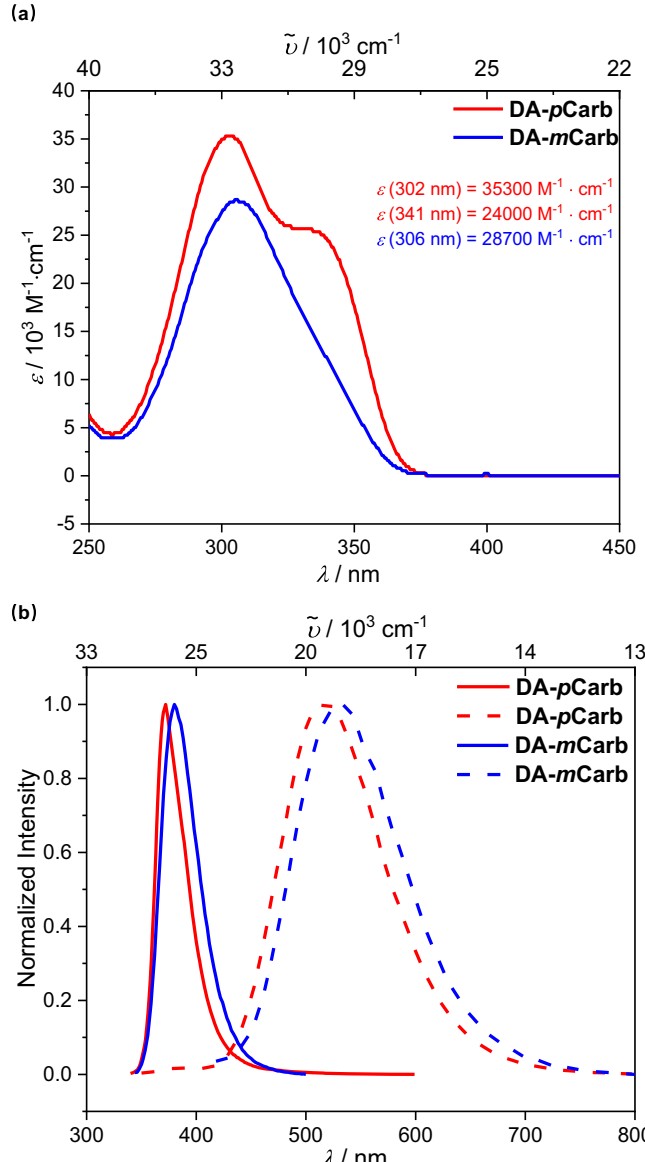

**(b)**

**Fig. 2 | Photophysical properties of DA-*p*Carb and DA-*m*Carb.** Absorption (**a**) and normalized emission (**b**) spectra of **DA-*p*Carb** (excited at 337 nm) and **DA-*m*Carb** (excited at 310 nm) in hexane (solid line) and THF (dashed line) at room temperature.

Because the band position and the 00-energy are difficult to estimate from the absorption spectra, we also measured the emission spectra of the four dyads in solvents of different polarities. The emission spectra of all four dyads are strongly solvatochromic (Supplementary Fig. 13). For **DA-*p*Carb** and **DA-*m*Carb** (Fig. 2b and Supplementary Fig. 13), the fluorescence spectra are broad and featureless in polar solvents, while the bands in the spectra are much sharper in hexane solutions. The Stokes shifts change from 2400 and 2900 cm$^{-1}$ in hexane to 15,100 and 13,700 cm$^{-1}$ in acetonitrile (Fig. 3, Supplementary Fig. 13), respectively, indicating the relaxed S$_1$ state has a larger dipole moment than the ground state. Determining the 00-energies (Supplementary Tab. 1) using a tangent to the high energy flank gives similar values of 28,100 cm$^{-1}$ in hexane for **DA-*p*Carb** and **DA-*m*Carb**, and 23,000 cm$^{-1}$ and 22,300 cm$^{-1}$ in THF for **DA-*p*Carb** and **DA-*m*Carb** respectively. These values indicate the formation of a relaxed charge transfer state in polar solvents. A fit to the emission spectra (Supplementary Fig. 14) of a large series of solvents with

varying polarity yielded a more accurate $\Delta G^{00}$ energy value (the difference in the free energy between the diabatic ground state and the excited state) as well as other reorganization parameters showing that both $\Delta G^{00}$ and the solvent reorganization energy $\lambda_o$ vary linearly with solvent polarity $f(\varepsilon)$-0.5 $f(n^2)$[65].

The solvatochromic effects were further investigated with Lippert-Mataga plots[66–68] (Fig. 3, Supplementary Tab. 2–5), which describe the difference of Stokes shifts ($\Delta\tilde{v}$) versus orientation polarizability ($\Delta f = f(\varepsilon)$-$f(n^2)$) in solvents of different polarities. The well-fitted linear correlation together with the large slopes confirms the ICT nature of the emissive S$_1$ state of both carboranes. The slope of **DA-*p*Carb** is about twice that of **DA-*p*Benz**, indicating a much stronger charge transfer in the former. The dipole moment changes ($\Delta\mu$) between the excited state and the ground state can be estimated using the effective radius of the spherical Onsager cavity of 7.07 Å and is about 44.9 D for **DA-*p*Carb**. Taking the dipole moment in the ground state ($\mu_g$) to be -1.06 D from DFT calculations (*vide infra*), we estimate the dipole moment of the relaxed S$_1$ state ($\mu_e$) to be at *ca.* 44 D, which corresponds to 86% charge-separation referring to the distance between nitrogen and the three-coordinate boron atom in the excited state. These values are similar to **DA-*m*Carb** ($\mu_g$ = 4.36 D, $\Delta\mu$ = 30.6 D) but almost twice that of **DA-*p*Benz** ($\mu_g$ = 2.31 D, $\mu_e$ = 18.4 D, 38% charge separation) and **DA-*m*Benz** ($\mu_g$ = 1.29 D, $\Delta\mu$ = 14.86 D).

The fluorescence quantum yields ($\Phi_f$) of the carborane-bridged compounds are strongly dependent on the polarity of the solvent, while the $\Phi_f$ values of **DA-*p*Benz** and **DA-*m*Benz** are not. The fluorescence quantum yield of **DA-*p*Carb** in THF is 64%, which is higher than the *meta*-carborane (35%). However, the quantum yield of **DA-*m*Carb** (5%) is similar to that of **DA-*p*Carb** (4%) in hexane. This observation is quite unusual for fluorescence from ICT states. In most donor-acceptor compounds, the ICT state is stabilized by polar solvents and the charges become more localized at the respective moieties. The latter usually leads to a smaller dipole strengths and thus a slow radiative decay rate, while the smaller vertical S$_1$ ← S$_0$ gap in more polar solvents leads to faster nonradiative deactivation, and consequentially to a lower quantum yield[69,70]. The trend of quantum yield vs. $\Delta G^{00}$ is given in the Supplementary Fig. 15 for both carboranes.

To obtain more insight into the emission dynamics, time-resolved fluorescence measurements were performed by time-correlated single photon counting (TCSPC) (Supplementary Fig. 16). The fluorescence lifetime ($\tau$) of carborane-bridged compounds in hexane are below the time resolution of our TCSPC setup (IRF = 0.9 ns). Indeed, transient absorption measurements with fs time resolution (*vide infra*) revealed the lifetime of the S$_1$ state in hexane to be only 0.18 ns (**DA-*p*Carb**) and 0.76 ns (**DA-*m*Carb**) which also explains the low fluorescence quantum yields. However, the fluorescence lifetimes of **DA-*p*Carb** (Table 2) in THF (24.2 ns) and DCM (20.4 ns) are two orders of magnitude longer than that in hexane, while lifetimes of **DA-*p*Benz** are little affected by solvent polarity (see Table 2). The radiative decay rate constant ($k_r$) of **DA-*p*Carb** in THF (2.6 ×10$^7$ s$^{-1}$) is an order of magnitude smaller than that in hexane (2.2 ×10$^8$ s$^{-1}$), while $k_r$ of **DA-*p*Benz** in hexane is very similar to that in THF. The fast nonradiative decay rate (5.3 ×10$^9$ s$^{-1}$) of **DA-*p*Carb** in hexane, which is 400 times higher than that in THF, is responsible for its low quantum yield. The squared transition dipole moment $\mu_{fl}^2$ ( = dipole strength) of **DA-*p*Carb** in hexane (17.5 D$^2$) was determined from the radiative rate constant by the Strickler-Berg equation[62] and is slightly larger than that of the absorption. On the other hand, $\mu_{fl}^2$ of **DA-*p*Carb** in THF is about one fourth of $\mu_{abs}^2$ indicating weaker orbital overlap between donor and acceptor moieties in the more polar solvent. In contrast, for **DA-*p*Benz** in both solvents $\mu_{fl}^2$ is larger than $\mu_{abs}^2$ (Table 2)[62–64,71].

The comparison of dipole strengths of absorption and emission for **DA-*p*Carb** indicates a more local excitation in the non-polar solvent but a CT character in the polar solvent where reduced orbital overlap leads to a smaller dipole strength for the emission. The increasing

**Table 2 | Photophysical data of DA-*p*Carb, DA-*m*Carb, DA-*p*Benz and DA-*m*Benz in hexane and in THF at room temperature**

|  | solvent | $\lambda_{abs}$ / nm ($\varepsilon^a$ / $10^4 M^{-1} cm^{-1}$) | $\lambda_{em}^b$ / nm | $\Delta \tilde{v}^c$ / cm$^{-1}$ | $\Phi^d$ | $\tau^e$ / ns | $k_r^h$ / $10^7 s^{-1}$ | $k_{nr}^i$ / $10^7 s^{-1}$ | $\mu_{abs}^2$ / D$^2$ | $\mu_{fl}^2$ / D$^2$ |
|---|---|---|---|---|---|---|---|---|---|---|
| **DA-*p*Carb** | hexane | 341 (2.40), 302 (3.53) | 372 | 2400 | 0.04 | 0.18 $^f$ | 22.2 | 533 | 14.7 | 17.5 |
|  | THF | 343, 302 | 515 | 9800 | 0.64 | 24.2 $^g$ | 2.6 | 1.5 | 19.6 | 5.4 |
| **DA-*m*Carb** | hexane | 342 (0.21), 306 (2.87) | 380 | 2900 | 0.03 | 0.76 $^f$ | 3.8 | 128 | 0.76 $^f$ | 3.2 |
|  | THF | 343, 307 | 530 | 10300 | 0.35 | 40.1 | 0.88 | 1.6 | 1.15 $^f$ | 2.0 |
| **DA-*p*Benz** | hexane | 384 (3.68), 340 (2.79), 275 (2.03) | 429 | 2700 | 0.89 | 2.03 | 44 | 5.2 | 31.8 | 54.9 |
|  | THF | 387, 341, 286 | 510 | 6200 | 0.88 | 4.0 | 23 | 2.8 | 27.9 | 43.3 |
| **DA-*m*Benz** | hexane | 326 (3.35), 296 (1.83) | 419 | 6800 | 0.31 | 5.1 | 6.1 | 13.6 | 22.5 | 6.9 |
|  | THF | 327, 294 | 513 | 11000 | 0.28 | 19.5 | 1.4 | 3.6 | 31.9 | 2.8 |

$^a$The molar extinction coefficient were determined by the absorbance of **DA-*p*Carb**, **DA-*m*Carb**, **DA-*p*Benz** and **DA-*m*Benz** in hexane at different concentrations (Supplementary Fig. 17);
$^b$Excited at 337 nm (**DA-*p*Carb**), 290 nm (**DA-*m*Carb**), 300 nm (**DA-*p*Benz**), 320 nm (**DA-*m*Benz**);
$^c$Stokes shift;
$^d$absolute quantum yields under argon atmosphere;
$^e$ fluorescence lifetimes measured by TCSPC;
$^f$ life-time from transient absorption measurement;
$^g$ measured under argon atmosphere;
$^h$ radiative decay rate constant $k_r = \Phi / \tau$;
$^i$ nonradiative decay rate constant $k_{nr} = (1- \Phi) / \tau$. $^f$ These values are afflicted with major inaccuracies as band integration is difficult because of strong band overlap.

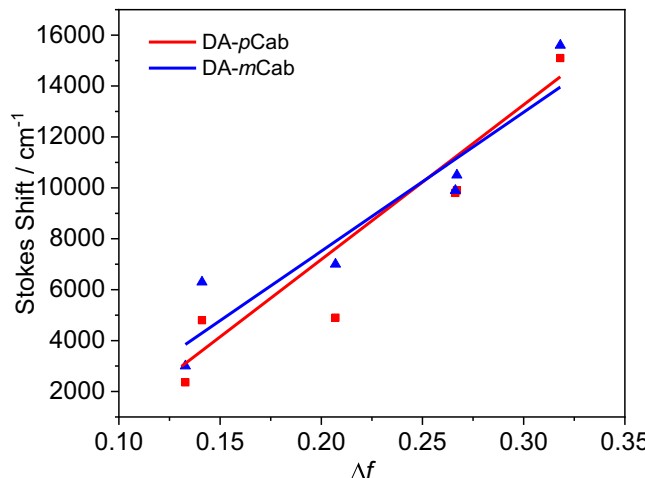

**Fig. 3 | Lippert-Mataga plots for emission of DA-*p*Carb (red) and DA-*m*Carb (blue).** Solvents used are from left to right: hexane, toluene, dibutyl ether, THF, DCM, and MeCN, corresponding to the squares/triangles in the picture.

fluorescence quantum yield in the more polar solvent is, at first sight, contradictory with common observations and the Engelman-Jortner[72] gap rule where one would expect a higher nonradiative rate constant for the stabilized CT state in a more polar solvent. Closer investigation of the nonradiative rate constant in an extended series of solvents (Supplementary Tab. 6, Supplementary Fig. 15) indeed manifests the trend that log$k_{nr}$ linearly increases with growing $\Delta G^{00}$ of the CT state. This trend which contradicts the Marcus inverted region behavior and the gap rule, is caused by an increasing degree of local excited state admixture and decreasing CT character in the emitting S$_1$ state. While in nonpolar solvents the CT character is less pronounced, it is dominating in more polar solvents but never reaches a full charge separation (*vide supra*). Indeed, Gould and Farid[73] found such a behavior in a series of donor-acceptor exciplexes if the amount of charge transfer is less than *ca*. 80%.

The room- and low-temperature emissions of **DA-*m*Carb** and **DA-*p*Carb** in methylcyclohexane were also measured (Supplementary Fig. 18–20). At 77 K, the phosphorescent bands around 450 nm in both compounds with lifetimes of *ca.* 600 ms are observed along with the fluorescence bands largely unchanged from the fluorescence bands

observed at room temperature. The emission spectra and lifetimes of all four compounds in PMMA and solid-state were also determined (Supplementary Fig. 21–23). These photophysical observations are typical of compounds with triphenylamine and BMes$_2$ substituents[74].

**Transient Absorption Spectroscopy**
To gain more insight into the excited state dynamics of both donor-acceptor carborane compounds, femtosecond pump-probe transient absorption spectroscopy (Fig. 4 and Supplementary Fig. 24) measurements in both hexane and THF were performed. The transient map was globally analyzed by a kinetic model involving a sequence of exponential functions. The derived evolution-associated difference spectra (EADS) revealed different behaviors of **DA-*p*Carb, DA-*p*Benz, DA-*m*Carb**, and **DA-*m*Benz** in the two solvents. For **DA-*p*Carb** in hexane, the first EADS with $\tau = 10$ ps shows two excited state absorption (ESA) bands (black line, Fig. 4a), one between 350-420 nm and the other one between 600-720 nm, both of which are typical absorption bands of triarylamine radical cations and arylBMes$_2$ radical anions[60,75,76]. This observation confirms the ICT character of the primarily populated S$_1$ state between the amino group and the BMes$_2$ moiety, and thus, an optically induced charge transfer where the ICT state is directly populated by absorption of light. There is a negative band at around 370 nm, which corresponds to the stimulated emission (SE) from this ICT state of **DA-*p*Carb**. Vibrational cooling/solvation was accomplished within 10 ps, giving the relaxed emissive ICT (= S$_1$) state with a lifetime of 0.18 ns (red line, Fig. 4a). The ESA spectrum shows almost no changes during this relaxation process, indicating the solvation effects in hexane are weak as expected for such a nonpolar solvent. In THF, the ESA signals are very similar to those in hexane, suggesting an ICT character of **DA-*p*Carb** in both solvents. Vibrational cooling with $\tau = 2.2$ ps in THF is accompanied by a small hypsochromic shift of the ESA band of the first EADS at 680 nm, confirming the strong solvation effect of the CT state in polar solvents[57]. The cold S$_1$ state has a lifetime of 20.5 ns, which agrees with the fluorescence lifetime of **DA-*p*Carb** in THF. Even more significant is the fading of the SE around 410–510 nm. However, because of the strong overlap of ESA and SE, it is difficult to extract specific parameters for the dynamic Stokes shift. Thus, the solvation process in THF was also investigated by broadband fluorescence upconversion spectroscopy (FLUPS) measurements, which show that the fluorescence shifts spectrally from 490 nm to 530 nm with time which is finished within 10 ps (Fig. 4f). The shortest component of the global deconvolution (Supplementary Fig. 24)

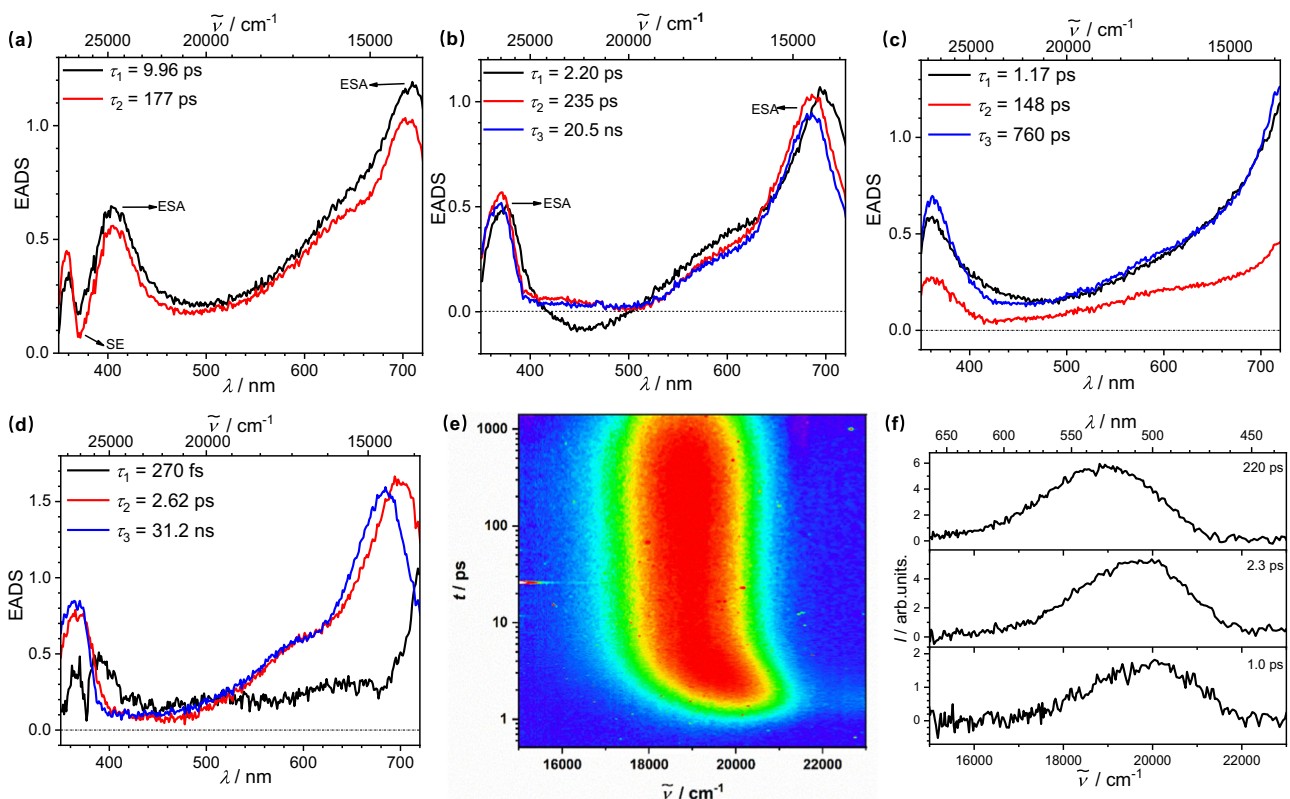

**Fig. 4 | Transient spectroscopy of DA-*p*Carb and DA-*m*Carb.** EADS and lifetimes from a global deconvolution of the transient absorption spectra of **DA-*p*Carb** in hexane (**a**) and THF (**b**), **DA-*m*Carb** in hexane (**c**) and THF (**d**), at room temperature excited at 340 nm; FLUPS emission map of **DA-*p*Carb** in THF excited at 340 nm (**e**) and fluorescence spectra (**f**) at indicated delay times.

possesses a time constant of 0.1 ps which is in good agreement with the shortest solvent correlation time of THF[77]. For **DA-*m*Carb** in hexane, the EADS only shows one ESA signal of triarylamine radical cation, and vibrational cooling with $\tau = 1.17$ ps giving a relaxed emissive state with a lifetime of 0.76 ns. In THF, vibrational cooling was finished within 270 fs, then the signal of anionic BMes$_2$ radical moiety was detected which is also accompanied by a hypsochromic shift. The cold $S_1$ state has a lifetime of 31.2 ns, indicating a slower radiative decay of the *meta*-carborane derivative.

For **DA-*p*Benz** in hexane, the EADS also shows two prominent ESA signals (*ca.* 480 nm and 610 nm), ground state bleaching (GSB) at 380 nm and SE at 430 nm. While the ESA and GSB signals are invariant with time, the SE signal shifts weakly with $\tau = 2.6$ ps (Supplementary Fig. 24). In THF, the first EADS with $\tau = 250$ fs is very similar to the one in hexane. A series of four EADS follows, which are not associated with different states but serve to model a continuous vibrational cooling/ solvent relaxation process expressed by a dynamic Stokes shift. In that way the two ESA signals shift from 480/620 nm to 450/570 nm and the SE from 430 nm to 510 nm. The latter is in very good agreement with the steady-state fluorescence maximum in THF, and the former with the one in hexane. The longest lifetime ($\tau = 3.4$ ns) is again in agreement with the fluorescence lifetime from the TCSPC measurement. Thus, in THF, the primarily populated state has the same characteristics as the one in hexane but unlike that in hexane, it displays a continuous combined molecular/solvent relaxation. The latter is essentially complete within *ca.* 3 ps. The two principal time constants of $\tau = 0.25$ and 0.73 ps are in very good agreement with the solvent relaxation times of THF[77].

## Density functional theory (DFT) studies

DFT calculations at B3LYP/6-31 G* in the gas phase were carried out to understand the interaction between the carborane and its substituents

(Fig. 5). We used CAM-B3LYP/6-31 G* for time-dependent DFT (TD-DFT) to identify likely transitions in the predicted absorption spectra but the excited states were heavily mixed with no obvious intermolecular charge transfer (ICT) state. B3LYP/6-31 G* on the other hand for TD-DFT provided reasonable results with a clear ICT character of the $S_1$ state. In both carboranes, the HOMO is mainly localized on the triphenylamine moiety with little contribution from the carborane bridge. While the LUMO of **DA-*p*Benz** and **DA-*m*Benz** is mainly located on the boron center and the biphenylene bridge (see SI), the LUMO of **DA-*p*Carb** and **DA-*m*Carb** is delocalized over the BMes$_2$ moiety and the entire carborane cage (Supplementary Fig. 25, 26). NAO (natural atomic orbitals) analyses were carried out to describe the fragment contributions to the frontier orbitals (Supplementary Tab. 7, 8)[78]. In the carborane derivatives, the LUMO consists of the empty boron $p_z$ orbital of the BMes$_2$ group (60% for **DA-*p*Carb** and 62% for **DA-*m*Carb**) and cluster orbitals (15% for *p*-carborane and 14% for *m*-carborane), confirming that the carborane cluster is significantly involved in the electronic structure. There is overlap between HOMO and LUMO in both compounds (Supplementary Fig. 25, 26), which is calculated at 0.18 in **DA-*p*Carb** and 0.11 in **DA-*m*Carb** where a complete overlap would be at 1.

Because of the spherical structure of the carborane bridge, we anticipated that the rotation of the phenylene ring of the triphenylamine might possess a very shallow potential. To get more insight how the rotation influences the conjugation properties of the carboranes, we rotated the phenylene group of **DA-*p*Carb** stepwise by 10° (torsional angle of C33-C32-B23-C1) and optimized all other structural parameters. The resulting potential is depicted in Fig. 6a and indeed shows a quite shallow potential of less than 60 cm$^{-1}$, which can easily be overcome by thermal energy at room temperature. The minimum structure refers to a parallel orientation of the amino-phenylene ring with the trigonal boron coordination plane

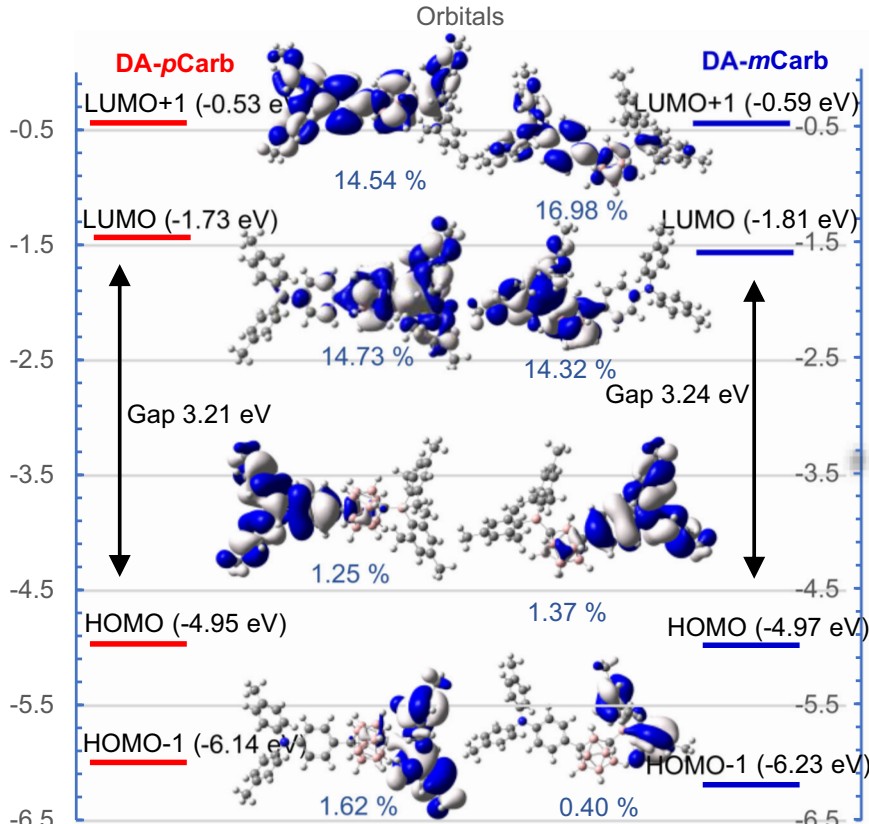

**Fig. 5 | Frontier molecular orbitals of DA-*p*Carb and DA-*m*Carb (B3LYP/6-31 G*).** Blue and white indicate the different signs of the wave function. The percentage values are the carborane-contributions to the molecular orbitals.

whereby the relative orientation of the nitrogen and boron $p_z$ orbitals is 39.8°. Taking this geometry as a start, we also optimized the $S_1$ state where this angle widens to 55.5°. On the other hand, the mesityl rings possess a somewhat smaller torsional angle with the trigonal nitrogen plane of 31° in the $S_1$ state compared to 45° in the $S_0$ state. We also calculated the $S_1 \leftarrow S_0$ transitions at B3LYP/6-31 G* for each fixed torsional step. While the overall transition energy is 23500 cm$^{-1}$ (425 nm), which is too low in comparison with experiment, the variation with the torsional angle shows a clear trend with a maximum transition energy for the minimum of the $S_0$ potential and a minimal transition energy for the maximum potential. However, the absolute variation is with 80 cm$^{-1}$ extremely small and would hardly be visible experimentally. A more important point is the variation of the oscillator strength, which goes from almost zero at the maximum to 0.15 at the minimum of the of the $S_0$ potential. Thus, orbital overlap between the phenylene ring and the boron center via the carborane cage clearly has a strong impact on the oscillator strength (Fig. 6, Supplementary Fig. 27, 28). For **DA-*m*Carb**, the phenylene plane and the B trigonal plane also are coplanar in the optimized global minimum structure. There is a similar potential barrier of *ca.* 90 cm$^{-1}$ for the rotation of the triarylamine in **DA-*m*Carb** as in **DA-*p*Carb** (Fig. 6). In contrast to **DA-*p*Carb** the $S_1 \leftarrow S_0$ transition energy is minimal in the potential minimum. However, the maximum oscillator strength is much smaller ($f = 0.03$) in the global minimum structure where the triarylamine phenylene ring and the boron $p_z$ orbitals are coplanar. This smaller oscillator strength is in agreement with the lower transition dipole strength of **DA-*m*Carb** (Fig. 2).

Both **DA-*m*Carb** and **DA-*p*Carb** show torsional-angle-dependent oscillator strengths, despite the spherical shape of the carborane. This reveals that orbital overlap plays a strong role in the electronic

transmission, that is, the dihedral angle of N and B plane has a vital influence for the electron communication through the carboranes. The HOMO and LUMO (Fig. 5, Supplementary Fig. 25, 26) at the optimized geometry show some overlap in the carborane region. The transmission of orbital overlap through the *meta*-carborane is weaker than for *para*-carborane as is apparent by the smaller oscillator strengths in the former. Both MOs are clearly π-orbitals and the twisting of the groups reduces this overlap and, consequently, the oscillator strength. The effect of carborane orbital contribution to the transmission of charge in the ICT process of **DA-*p*Carb** was estimated by a TD-DFT calculation in which the carborane cluster atoms in the global minimum structure were omitted and hydrogen atoms were added to the phenylene and boron. This calculation gave an ICT energy, which is 1500 cm$^{-1}$ higher but possesses zero oscillator strength instead of 0.15 for **DA-*p*Carb**. The contribution of ICT (LUMO ← HOMO) transition to the $S_1 \leftarrow S_0$ transition is nearly 100% in all four compounds (Supplementary Tab. 9–12, Supplementary Fig. 29–33). The calculated oscillator strength (0.58) in **DA-*p*Benz** is much higher than in **DA-*p*Carb** and the same is true for **DA-*m*Carb** and **DA-*m*Benz**, which matches well with the comparatively large transition dipole strength that was estimated from the experimental absorption spectrum (Table 2 and Supplementary Tab. 6).

## Discussion

In conclusion, we have synthesized the donor-acceptor systems **DA-*p*Carb** and **DA-*m*Carb** and studied the communication through carboranes by combining absorption spectroscopy, steady-state and time-resolved fluorescence spectroscopy, fs- transient absorption, FLUPS, cyclic voltammetry, DFT and TD-DFT calculations. Unexpectedly, the fluorescence quantum yield increases with the solvent

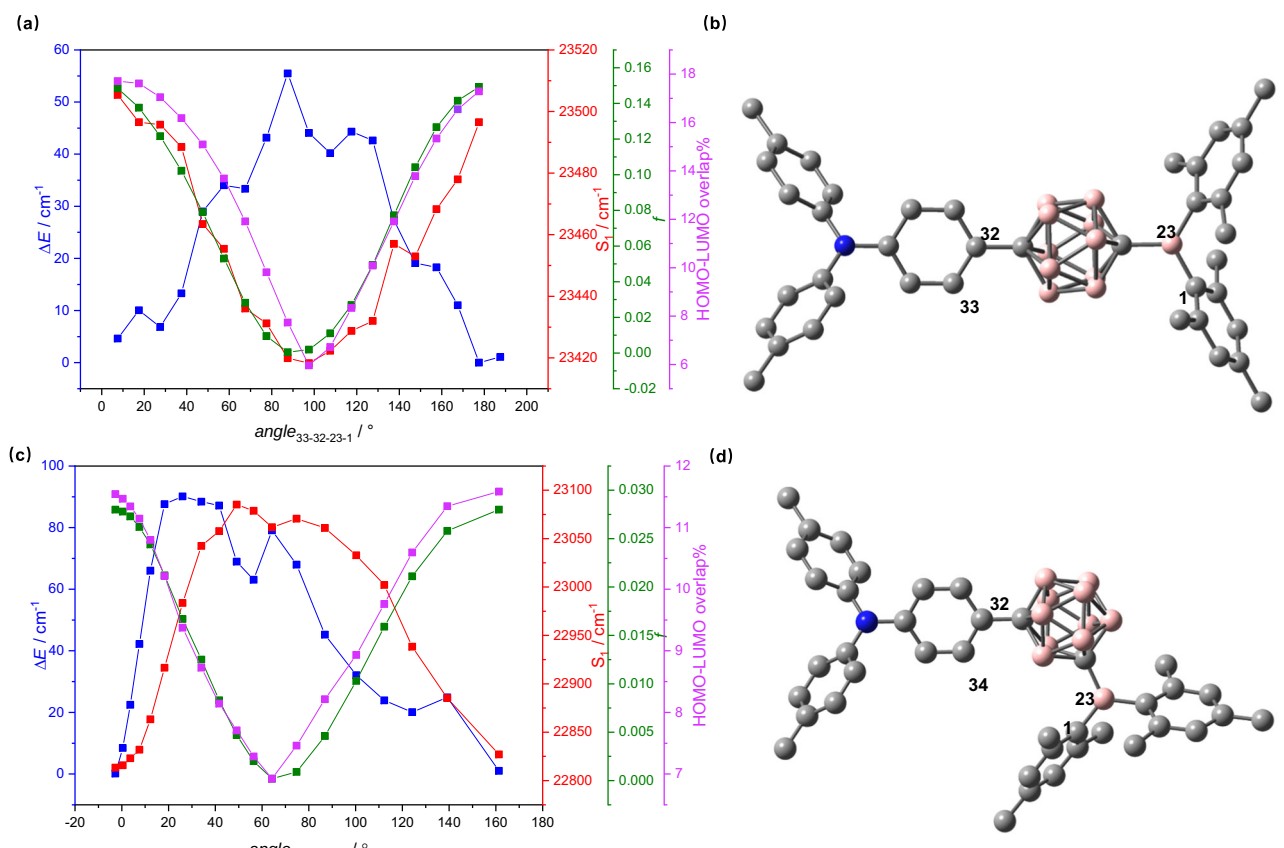

**Fig. 6 | Conformer analyses.** Relative ground state energy depending on the phenylene-borane-plane torsional angle (blue) and absolute $S_1 \leftarrow S_0$ transition energies (red), oscillator strength $f$ (green) and HOMO-LUMO overlap % (purple) in the gas phase of **a) DA-$p$Carb** and **c) DA-$m$Carb**). Optimized global minimum structures of **b) DA-$p$Carb** and **d) DA-$m$Carb**; blue spheres, pink spheres and gray spheres corresponding to the nitrogen, boron, and carbon atoms, respectively.

polarity which is caused by a decrease of nonradiative rate constant tuned by the amount of local excitation to the CT character of the $S_1$ state. The dipole moment of the relaxed $S_1$ state of both carborane derivatives is much larger than that of analogous phenylene-bridged derivatives, indicating a more pronounced charge transfer through the carborane bridge than in the case of phenylene. Femto-second transition absorption measurements and FLUPS of **DA-$p$Carb** reveals the spontaneous formation of the charge transfer state with excitation. Thus, the lowest energy absorption and emission band of **DA-$p$Carb** correspond to the ICT transition between the amine and the BMes$_2$ moieties through the $p$-carborane cage with considerable transition dipole strength. A less pronounced optically induced process was found in **DA-$m$Carb**. TD-DFT calculations show that the $S_1 \leftarrow S_0$ transitions of both carborane derivatives are predominantly LUMO ← HOMO transitions that correspond to the migration of electrons from the triphenylamine to the BMes$_2$ group through the carborane bridge. This transition depends on orbital overlap because it is influenced by the dihedral angle of the trigonal N and B plane of the carborane compounds. Maximum oscillator strengths were obtained when the phenylene ring is coplanar with the trigonal boron plane when the HOMO - LUMO overlaps are at maximum. Both experimental and theoretical results confirm the charge transfer through $m$- and $p$- carborane bridges, indicating considerable conjugation between the 3-D carborane bridge and the adjacent donor and acceptor 2-D π-systems. Concerning the fluorescence properties (energy, quantum yield, lifetime), *meta*- and *para*-substituted carboranes and the *meta*-phenylene analog are all similar. Our results confirm the potential of developing carborane-bridged fluorescent materials with high quantum yields in polar media.

## Methods
### Materials

For $^{11}$B NMR, $^{11}$B{$^{1}$H} NMR, $^{1}$H NMR, $^{1}$H{$^{11}$B} NMR, $^{13}$C{$^{1}$H} NMR, and high-resolution mass spectrometry of compounds in this manuscript and details of the synthetic procedures, see Supplementary Information.

## Data availability
All data supporting the findings of this study, including experimental procedures and compounds characterization, the relevent measurement methods, computational details, are avaliable within the paper and Supplementary Information. The coordinates of the optimized structures are present as source data. All data are available from the corresponding author upon request. Source data are provided with this paper.

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

## Acknowledgements

We are grateful for generous financial support by the Nature Science Foundation of China (Grant No. 62174137), the Key Research and Development Projects of Shaanxi Province (Grant No. 2020GXLH-Z-022), Ningbo Natural Science Foundation (Grant No. 2021J044, 2023J038), the Fundamental Research Funds for the Central Universities, and Northwestern Polytechnical University. L. W. acknowledges generous financial support from the China Scholarship Council. C.L. thanks the Bavarian State Ministry of Education, Culture, Science, and the Arts for funding this work by the SolTech initiative. We gratefully thank Prof. Dr. Dr. h.c. Todd B. Marder for selfless support during the whole project, we also thank the kind help from Dr. Florian Rauch.

## Author contributions

L.J., C.L., M.A.F. and M.F. conceived and directed the project. L.W. and F.P. synthesized and characterized the compounds reported in this paper. L.W., A.S. and P.M. did the photophysical measurements. L.W., M.H. J.S. did the theoretical studies. M.M. did the electrochemistry. L.W., T.N., R.B. did the boron-related NMR studies and M.A.F. analyzed the data.

## Competing interests

The authors declare no competing interests.
