## [Peer Review File · Nature Communications]

Optically Induced Charge-Transfer in Donor-Acceptor Substituted p- and m- C₂B₁₀H₁₂ CarboranesReviewers' Comments:

Reviewer #1:

Remarks to the Author:

I will not waste time reading this paper and carefully considering it until the Authors provide expert NMR data for the compounds. I am an expert at carborane chemistry and based on what they have provided I cannot assign the compounds or judge their purity. There are too many unidentified impurities in the provided NMR data and their labeling is incorrect on some spectra. The compound numbering style is confusing and not homogenous in the SI. Please address the above and following comments if you would like me to consider this manuscript, as in its current form it is not publishable anywhere.

- 1) Provide cleaner ^1H , $\{^1\text{H}\}^{13}\text{C}$, spectra for all compounds. If the compounds are misidentified or are not clean all the UV spectra and other physical property data is unreliable as it can be adulterated by impurities. You provide no ^{11}B NMR data, please provide both ^{11}B and $\{^1\text{H}\}\{^{11}\text{B}\}$ as well as $\{^{11}\text{B}\}^1\text{H}$ and fix your incorrect labels of the ^{13}C spectra. The ^{13}C is clearly ^1H decoupled but you label it as ^{13}C . And assign peaks to the best of your ability....
- 2) Make your compound numbering consistent in the SI and provide an actual chem draw picture for the compounds above the data so the reader can easily analyze the data without referring back to the manuscript itself. Turn your NMR spectra so 90° so the reader doesn't have to do this themselves when reading through the SI.
- 3) None of the work of Lavallo is cited related to cross conjugation between cage and exosubstituents. There are several papers with the CB11 system after cyclization reactions occur at cage.

Reviewer #2:

Remarks to the Author:

The authors, L. Wu, C. Lambert, L. Jia, et. al. reported on the communication paper for two donor-acceptor substituted m- and p- carboranes, DA-mCarb and DA-pCarb, and studied their photophysical and electrochemical properties and further, comparison of their characteristics to phenylene-bridged analogue. The concept that the m- and p-carboranes could be replaced to the position of benzene ring and they could show the intriguing photophysical properties looks a little fancy and the authors tried to demonstrate these analyses in detail. Furthermore, the description for the Introduction section well focused on this research scoop and the added reference papers look proper; however, on this occasion the reviewer does not feel that this work is suitable for the journal Nature Communication because it does not meet the very high significance and general interest standards of the journal. The reviewer thinks that this will be of significant interest to the organic/inorganic materials and their engineering communities chemistry, similar materials have been previously reported and there was insufficient demonstration that the design and photophysical insight resulted in improved performance compared to state of the art related materials. Therefore, this study lacks the demonstrated impact and broad appeal necessary to interest our general chemistry readership and your manuscript has been rejected from Nature Communication. The reviewer does consider that it would be of significant interest to an inorganic/organometallic-chemistry readership and therefore I would strongly encourage to transfer this study to specific journal.

Before transfer, there are the simple parts which the authors should add to the manuscript and revise as the following:

1. Fundamentally, it is possible that different kinds of photophysical data (quantum efficiency, radiative decay, and so on) for carborane and phenylene derivatives compared each other (LE and ICT-based emission, respectively)? The authors should consider the paradox.

2. The reviewer thinks that the contents of Abstract look not matched to the overall contents of the manuscript. It looks like the author try to emphasize the possibility or outstanding performance for carborane replacement; however, the manuscript just includes the photophysical characteristics affected by polarity and dipole strength. The authors should correctly point out what you want to show and emphasize.

3. It seems like it will be better that the authors provide the photophysical data of DA-mCarb, DA-pCarb, DA-mBenz, DA-pBenz in rigid states (such as solution at 77 K, film (solid) or crystalline).

4. Page 12 middle: to support the sentence 'Thus, orbital overlap between the phenylene ring and the boron center as transmitted by the carborane cage clearly has a strong impact on the oscillator strength.', the authors should present the figure of MO or add the overlap% to Figure

5. Figure 3a: The dash-line should be matched to zero-level.

6. Figure 3b: Please check the blue and black lines. The reviewer thinks that these should be switched.

8. Figure S12: Please revise the data in the order of polarity (hexane-toluene-THF-DCM)

9. Table 2: DA-mCarb hexane data need the footnote (f).

10. Please unify the x-axis unit to one of them, wavelength or wavenumber in all the Figures.

Reviewer #3:

Remarks to the Author:

The science presented here is fine and is worthy of publication. However, I see that the authors failed to cite some relevant literature published in books and journals. This must be corrected before allowing publication of this work!

If the authors type "books on Boron Compounds and their Applications in Materials Science and Medicine" in their web browser, they will see many citations including many books published in recent years (<https://www.ncbi.nlm.nih.gov/pmc/articles/PMC7071021/>;
<https://www.worldscientific.com/worldscibooks/10.1142/q0130#t=aboutBook>;
<https://www.wiley.com/en-us/Boron+Based+Compounds:+Potential+and+Emerging+Applications+in+Medicine-p-9781119275558>; <https://www.hindawi.com/journals/jchem/si/919356/>;
<https://www.worldscientific.com/worldscibooks/10.1142/8056>;
<https://www.springer.com/us/book/9780306445675>;
<https://www.springer.com/us/book/9783642313332>; <https://www.crcpress.com/Boron-Science-New-Technologies-and-Applications/Hosmane/p/book/9781439826621>;
https://hochitw.com/index_down.php?openCatIDfor3=&openCatID=0&CAhs=&firestpageset=1&ISPID=14909&IIBig=71&sele=shopbig_dm_right; <https://www.worldscientific.com/doi/10.1142/q0130-vol4>). Without the citation of these books, along with relevant journal publications, I consider this manuscript is "Incomplete"!

2. After appropriate major revision, the manuscript needs to be reviewed again before publication.

We thank all reviewers for their suggestions to improve our manuscript. We believe we have addressed all points in detail as listed below, and modified the manuscript and SI accordingly. We also included additional references as suggested by the reviewers. The modifications in the revised manuscript and SI are highlighted in yellow.

Reviewer #1

I will not waste time reading this paper and carefully considering it until the Authors provide expert NMR data for the compounds. I am an expert at carborane chemistry and based on what they have provided I cannot assign the compounds or judge their purity. There are too many unidentified impurities in the provided NMR data and there labeling is incorrect on some spectra. The compound numbering style is confusing and not homogenous in the SI. Please address the above and following comments if you would like me to consider this manuscript, as in its current form it is not publishable anywhere.

We believe we have addressed all raised issues in our revised version as detailed below.

1) Provide cleaner ^1H , $\{^1\text{H}\}^{13}\text{C}$, spectra for all compounds. If the compounds are misidentified or are not clean all the UV spectra and other physical property data is unreliable as it can be adulterated by impurities. You provide no ^{11}B NMR data, please provide both ^{11}B and $\{^1\text{H}\}^{11}\text{B}$ as well as $\{^{11}\text{B}\}^1\text{H}$ and fix your incorrect labels of the ^{13}C spectra. The ^{13}C is clearly ^1H decoupled but you label it as ^{13}C . And assign peaks to the best of your ability.

We thank the reviewer for the recommendation. The $^1\text{H}\{^{11}\text{B}\}$ NMR, ^{11}B NMR, and $^{11}\text{B}\{^1\text{H}\}$ NMR spectra of **D-pCarb**, **D-mCarb**, **DA-pCarb** and **DA-mCarb** have been added in "NMR Spectra" part in the supplementary Information (SI). We have also assigned all peaks where possible with the aid of ChemDraw figures containing compound numbers for the assignments in the SI.

2) Make your compound numbering consistent in the SI and provide an actual chem draw picture for the compounds above the data so the reader can easily analyze the data without referring back to the manuscript itself. Turn your NMR spectra so 90° so the reader doesn't have to do this themselves when reading through the SI.

We thank the reviewer for this constructive suggestion. We have added labeled ChemDraw figures of the corresponding compounds in the SI. We have also rotated all NMR spectra by 90° for better viewing.

3) None of the work of Lavallo is cited related to cross conjugation between cage and exosubstituents. There are several papers with the CB11 system after cyclization reactions occur at cage.

We thank the reviewer for pointing out some important references to add and have cited two references (Ref. 31 - 32) that are related to our study.

Reviewer #2:

The authors, L. Wu, C. Lambert, L. Jia, et. al. reported on the communication paper for two donor-acceptor substituted m- and p- carboranes, DA-mCarb and DA-pCarb, and studied their photophysical and electrochemical properties and further, comparison of their characteristics to phenylene-bridged analogue. The concept that the m- and p-carboranes could be replaced to the position of benzene ring and they could show the intriguing photophysical properties looks a little fancy and the authors tried to demonstrate these analysis in detail. Furthermore, the description for the Introduction section well focused on this research scoop and the added reference papers look proper; however, on this occasion the reviewer does not feel that this work is suitable for the journal Nature Communication because it does not meet the very high significance and general interest standards of the journal. The reviewer thinks that this will be of significant interest to the organic/inorganic materials and their engineering communities chemistry, similar materials have been previously reported and there was insufficient demonstration that the design and photophysical insight resulted in improved performance compared to state of the art related materials. Therefore, this study lacks the demonstrated impact and broad appeal necessary to interest our general chemistry readership and your manuscript has been rejected from Nature Communication. The reviewer does consider that it would be of significant interest to an inorganic/organometallic-chemistry readership and therefore I would strongly encourage to transfer this study to specific journal.

Before transfer, there are the simple parts which the authors should add to the manuscript and revise as the following:

The novelty of our manuscript is that we achieved the π -like communication through carborane linkers for the first time in the excited states, this conjugation is a very basic question and an important finding in chemistry. We believe it is of great interest to scientists working in organic, physical and theoretical chemistry of organic fluorescent materials. We have changed the phrase "optical functional materials" to "optical functional chromophores" in the abstract and introduction, as it implies in this reviewer's opinion that we are focusing on improved performances with new materials.

1) Fundamentally, it is possible that different kinds the photophysical data (quantum efficiency, radiative decay, and so on) for carborane and phenylene derivatives compared each other (LE and ICT-based emission, respectively)? The authors should consider the paradox.

We are not sure why this reviewer considers that our carborane derivatives give LE-based emissions. Both carborane and phenylene derivatives have ICT-based emissions as described in the manuscript.

2) The reviewer thinks that the contents of Abstract look not matched to the overall contents of the manuscript. It looks like the author try to emphasize the possibility or outstanding performance for carborane replacement; however, the manuscript just includes the

photophysical characteristics affected by polarity and dipole strength. The authors should correctly point out what you want to show and emphasize.

We have changed the phrase “optical functional materials” to “optical functional chromophores” in the abstract as it appears to mislead this reviewer’s view of our intriguing study as stated earlier. The emphasis of this work is the evidence of communication between donor and acceptor groups via the carborane linker which has not been demonstrated before. This comment is stated in the abstract.

3) It seems like it will be better that the authors provide the photophysical data of DA-mCarb, DA-pCarb, DA-mBenz, DA-pBenz in rigid states (such as solution at 77 K, film (solid) or crystalline).

We thank the reviewer for this suggestion. We have measured the room- and low-temperature emission of **DA-mCarb** and **DA-pCarb** in methylcyclohexane and added them to the SI (Figure S14-S16). The new bands around 450 nm in both compounds are phosphorescent, with dual-lifetimes of *ca.* 100 and 600 ms. The emission spectra of those compounds at 77 K contain new phosphorescence bands, while the fluorescence bands are largely unchanged from the fluorescence bands observed at room temperature. These observations are typical of triphenylamino- and -BMe₂ containing compounds (Reference 74: *Angew. Chem., Int. Ed.* **61**, e202206366 (2022)). We also measured the emission spectra of all four compounds in PMMA and solid-state and included them in the (Figure S17-S19).

We added related description in main text: “The room- and low-temperature emissions of **DA-mCarb** and **DA-pCarb** in methylcyclohexane were also measured (Figure S14-S16). At 77 K, the phosphorescent bands around 450 nm in both compounds with lifetimes of *ca.* 600 ms are observed along with the fluorescence bands largely unchanged from the fluorescence bands observed at room temperature. The emission spectra and lifetimes of all four compounds in PMMA and solid-state were also determined (Figure S17-S19). These photophysical observations are typical of compounds with triphenylamino- and -BMe₂ substituents.⁷¹”

4) Page 12 middle: to support the sentence ‘Thus, orbital overlap between the phenylene ring and the boron center as transmitted by the carborane cage clearly has a strong impact on the oscillator strength.’, the authors should present the figure of MO or add the overlap% to Figure

We thank the reviewer for this constructive and important suggestion. We have added the % overlaps to Figure 5 (purple axis), we also present the figure of MOs for intuitive reading (Figure S23 - S24, see also below).

Figure R1: Molecular orbitals of DA-*p*Carb in different orientations.

Figure R2: Molecular orbitals of DA-mCarb in different orientations.

5) Figure 3a: The dash-line should be matched to zero-level.

We agree. The dash-line is now matched to zero-level.

6) Figure 3b: Please check the blue and black lines. The reviewer thinks that these should be switched.

We thank the reviewer for this suggestion. We have checked these lines, the assignments in Figure 3b are correct and don't need to switch.

8) Figure S12: Please revise the data in the order of polarity (hexane-toluene-THF-DCM)

We thank the reviewer for this good point. We have revised the data accordingly.

9) Table 2: DA-mCarb hexane data need the footnote (f).

We agree. We have added the footnote (f) to the revised manuscript.

10) Please unify the x-axis unit to one of them, wavelength or wavenumber in all the Figures.

We have unified the x-axis unit of all pictures and used both wavelength and wavenumber in all the figures except Figures S3-S7, as the Gaussian fittings are all done in wavenumber units.

Reviewer #3:

The science presented here is fine and is worthy of publication. However, I see that the authors failed to cite some relevant literature published in books and journals. This must be corrected before allowing publication of this work!

We are pleased that this reviewer considers that our work is suitable for publication in Nature Communications.

If the authors type "books on Boron Compounds and their Applications in Materials Science and Medicine" in their web browser, they will see many citations including many books published in recent years (<https://www.ncbi.nlm.nih.gov/pmc/articles/PMC7071021/>; <https://www.worldscientific.com/worldscibooks/10.1142/q0130#t=aboutBook>; <https://www.wiley.com/en-us/Boron+Based+Compounds:+Potential+and+Emerging+Applications+in+Medicine-p-9781119275558>; <https://www.hindawi.com/journals/jchem/si/919356/>; <https://www.worldscientific.com/worldscibooks/10.1142/8056>; <https://www.springer.com/us/book/9780306445675>; <https://www.springer.com/us/book/9783642313332>; <https://www.crcpress.com/Boron-Science-New-Technologies-and-Applications/Hosmane/p/book/9781439826621>; https://hochitw.com/index_down.php?openCatIDfor3=&openCatID=0&CAhs=&firestpageset=1&ISPID=14909&IIBig=71&sele=shopbig_dm_right; <https://www.worldscientific.com/doi/10.1142/q0130-vol4>). Without the citation of these books, along with relevant journal publications, I consider this manuscript is "Incomplete"!

We thank the reviewer for these important references. We have cited relevant article/books in the revised version of manuscript (Ref.10 - 14).

Reviewers' Comments:

Reviewer #1:

Remarks to the Author:

I was reviewer 1 in round 1. The authors have addressed all of my concerns in an expert fashion and I have now carefully reviewed the manuscript. The SI is perfect and the compounds structures are properly assigned, therefore the optical data is valid. So to be clear the UV spectra and other physical property data is reliable. Ok so now onto the importance of the findings. As the authors point out and reviewer 2 seems to not be aware of there is a longstanding debate about cross conjugation between 3-D cages like this and 2-D pi systems. Some authors claim there is cross conjugation and sometimes aromaticity, and some theoreticians recently reported in Chem Sci that prior reports were mistaken. This study clearly demonstrates cross conjugation by showing how the physical properties of the molecules in question change upon rotation of the 2-D pi system with respect to the cages orbitals of pi symmetry. Additionally, the authors probe these effects in the excited state which certainly has not been done before, thus this new study is very novel. I am a bit perplexed by reviewer 2's negativity, and he is clearly not an expert in carboranes but perhaps a materials scientist or engineer. Essentially what this study does is provide physical evidence of fundamentally important and controversial structure and bonding intricacies that have implications in creating superior optical and electronic materials. If one can translate the stability of carboranes into competitive technologies for optical and electronic materials, this could lead to various technologies being more long lived and more resistant to decomposition in high energy environments. Therefore, I view this work as highly important as it provides the climbing gear to reach new pinnacles and it will most certainly be of very broad interest to various communities.

Reviewer #2:

Remarks to the Author:

The authors have addressed the reviewers' comments well and improved the manuscript a lot. I think the manuscript is suitable for publication in this case.

Reviewer #3:

Remarks to the Author:

The revised manuscript answered most of the questions raised by the reviewers and if other reviewers are happy with the revised version, then I have no objection to recommend this manuscript for publication in Nature Communications. Nonetheless, the authors could have used some English experts in China to improve their ENGLISH in the text so that this communication can be read worldwide and cite it accordingly. In any case, it is left to the authors and the other reviewers.